 

# GPER is a mechanoregulator of pancreatic stellate cells and the tumor microenvironment

Ernesto Cortes[1],[†], Muge Sarper[1],[†], Benjamin Robinson[1],[†], Dariusz Lachowski[1] (iD), Antonios Chronopoulos[1], Stephen D Thorpe[2],[*], David A Lee[2] (iD) & Armando E del Río Hernández[1],[**] (iD)

## Abstract

The mechanical properties of the tumor microenvironment are emerging as attractive targets for the development of therapies. Tamoxifen, an agonist of the G protein-coupled estrogen receptor (GPER), is widely used to treat estrogen-positive breast cancer. Here, we show that tamoxifen mechanically reprograms the tumor microenvironment through a newly identified GPER-mediated mechanism. Tamoxifen inhibits the myofibroblastic differentiation of pancreatic stellate cells (PSCs) in the tumor microenvironment of pancreatic cancer in an acto-myosin-dependent manner via RhoA-mediated contractility, YAP deactivation, and GPER signaling. This hampers the ability of PSCs to remodel the extracellular matrix and to promote cancer cell invasion. Tamoxifen also reduces the recruitment and polarization to the M2 phenotype of tumor-associated macrophages. Our results highlight GPER as a mechanical regulator of the tumor microenvironment that targets the three hallmarks of pancreatic cancer: desmoplasia, inflammation, and immune suppression. The well-established safety of tamoxifen in clinics may offer the possibility to redirect the singular focus of tamoxifen on the cancer cells to the greater tumor microenvironment and lead a new strategy of drug repurposing.

**Keywords** GPER; mechanotransduction; RhoA signaling; tamoxifen; tumor microenvironment

**Subject Categories** Cancer; Signal Transduction

See also: **E Cortes *et al*** (January 2019) and **M Pein & T Oskarsson** (January 2019)

## Introduction

Pancreatic ductal adenocarcinoma (PDAC) is one of the most lethal cancers, and its survival rate has not significantly changed in the past 40 years. The defining feature of PDAC is extensive tissue fibrosis or desmoplasia, which composes the majority of the stroma surrounding the tumor, and is characterized by an excessive deposition and cross-linking of extracellular matrix (ECM) proteins resulting in increased ECM rigidity [1]. This stiff stroma provides a unique microenvironment that hampers drug delivery and modulates pancreatic tumor behavior, including its ability to grow, metastasize, as well as resist chemotherapy [1–4]. Desmoplasia is causally linked with inflammation and has been recently described as a potential immune regulator in PDAC by impacting the infiltration of immune cells and thereby regulating tumor aggression [3,5–7].

The increasing, but still emerging, appreciation that the desmoplastic stroma is not a bystander in pancreatic cancer has underscored the relevance of pancreatic fibrosis as an attractive target for PDAC therapies. Several attempts to target the stroma have been carried out in the past [3,5,8,9]. A more subtle strategy of reprogramming stromal cells to render them quiescent seems to offer an attractive avenue for adjunct PDAC therapy [4,10]. Pancreatic stellate cells (PSCs) are the main cell type in the tumor microenvironment and drive the desmoplastic reaction in the pancreas. In the healthy pancreas, PSCs are quiescent, have abundant vitamin A cytoplasmic vesicles, and have low levels of ECM production. An integral feature of PSCs in PDAC is their transition to an activated state whereby they lose their cytoplasmic vesicles and adopt a myofibroblast-like contractile phenotype expressing high levels of alpha smooth muscle actin (αSMA) [2]. Thus, we hypothesized that pharmacological intervention that normalizes PSC mechanobiology may be of therapeutic value.

Tamoxifen is an anti-estrogen drug widely used in hormonal therapy for breast cancer, but it is also a G protein-coupled estrogen receptor (GPER) agonist [11,12]. Interestingly, tamoxifen decreases myofibroblast contractility and their ability to deform the underlying matrix [13], which suggests that tamoxifen may be a suitable agent to modulate PSC myofibroblast activation. Here, we investigated the role of GPER in the PDAC stroma and found that tamoxifen, acting through GPER, suppressed fibrosis and modulated immune response in PDAC mouse models. We also observed that tamoxifen reprogrammed PSCs via GPER to inhibit myofibroblastic differentiation and their capacity to remodel the stroma, which suppressed cancer cell invasion. Our results suggest that GPER is a mechanoregulator of stromal function that induces multiple changes in the tumor

1 Cellular and Molecular Biomechanics Laboratory, Department of Bioengineering, Imperial College London, London, UK
2 Institute of Bioengineering, School of Engineering and Materials Science, Queen Mary University of London, London, UK
   *Corresponding author. Tel: +44 20 7882 3602; E-mail: s.thorpe@qmul.ac.uk
   **Corresponding author. Tel: +44 20 7594 5187; E-mail: a.del-rio-hernandez@imperial.ac.uk
   †These authors contributed equally to this work

microenvironment to target desmoplasia, inflammation, and immune function in PDAC. Intriguingly, tamoxifen is widely used to induce the expression of specific phenotypes in conditional somatic mouse mutants (experimental mice for inducible gene knockouts) [14], and its administration may alter the biomechanical homeostasis and immune responses of the tissue under study. This highlights the need for caution in using these tamoxifen-inducible Cre mice models in the cases where long-term exposure to tamoxifen is needed to resemble chronic conditions, or when the output is assessed immediately after tamoxifen treatment.

## Results and Discussion

### Tamoxifen decreases fibrosis, macrophage recruitment, and macrophage polarization

To investigate the effect of tamoxifen in PDAC-associated desmoplasia, we used a genetically engineered mouse model of pancreatic cancer. Specifically, the KPC model (Pdx-1 Cre, Kras$^{G12D/+}$, p53$^{R172H/+}$), which is a well-validated and clinically relevant model that recapitulates PDAC progression in a known timescale [15]. Tumor-bearing KPC mice were randomized to three groups and were injected intraperitoneally (IP) with either: (i) vehicle [control hereafter], (ii) 2 mg, or (iii) 5 mg of tamoxifen daily (Fig 1A). The doses were selected according to previous work and to match the doses commonly used in tamoxifen-inducible gene knockout mice [16,17]. After the treatment, mice were culled and pancreatic tissues harvested and used for further analysis. To test how tamoxifen treatment impacts tissue mechanics, we used atomic force microscopy (AFM) to calculate an average of Young's modulus, which indicates tissue stiffness and mechanics (Appendix Fig S1). We observed a remarkable dose-dependent 10-fold decrease in tissue stiffness after tamoxifen treatment. Because collagen I (collagen hereafter) is the principal source of fibrotic matrix in PDAC [2], we used Sirius Red staining to evaluate the collagen density and organization in the pancreata of each mice group. The staining intensity gradually decreased twofold when tissues from mice were treated with 5 mg of tamoxifen with respect to control mice. The collagen fibers in control mice were also more aligned in comparison with treated mice (Fig 1B and Appendix Fig S1). The thickness of collagen fibers also decreased up to 40% in a dose-dependent manner in the 2 mg and 5 mg tamoxifen-treated mice (Appendix Fig S1). In pancreatic desmoplasia, higher deposition of collagen correlates with strong expression of αSMA by activated PSCs [18]. Interestingly, tamoxifen reduces renal fibrosis by decreasing ECM deposition and expression of αSMA in mouse models [19]. In agreement with this, we observed a gradual and significant 50% decrease in the percentage of αSMA positive cells (Fig 1C).

It has been shown that PSCs and macrophages mutually activate each other, and cytokines secreted by macrophages enhance the expression of αSMA in activated PSCs [20]. Indeed, PDAC tissues are highly infiltrated by tumor-associated macrophages (TAM), which predominantly carry the M2 phenotype (alternative activation by secretion of immune suppressor cytokines such as IL-10) [21,22]. Unlike their classically activated anti-tumor M1 counterparts, M2-type TAMs mask anti-tumor immunity and secrete several factors which persuade ECM secretion and remodeling, therefore promoting tumor-favoring microenvironment [23,24]. Certainly, M2-type infiltration is shown to be associated with invasion, metastasis, angiogenesis, lymphangiogenesis, lower overall survival and resistance to chemotherapy in pancreatic cancer [21,22,24–26]. Intriguingly, M2 TAM influx was shown to be significantly correlated with the stiffness of the tumor tissue [27]. Figure 1D shows that the percentage of macrophages (CD68 positive cells) in formalin-fixed paraffin-embedded pancreas tissues was reduced upon tamoxifen treatment, which is observed in both 2 mg and 5 mg treatment groups as compared to vehicle-only controls in a dose-dependent manner. To further characterize the polarization of tumor-associated macrophages, we used CD204 macrophage scavenger receptor as marker for the M2 phenotype on freshly frozen PDAC sections from KPC mice [21,22]. As shown in Fig 1E, the percentage of CD204-positive cells was significantly reduced by increased doses of tamoxifen compared to control mice. These data collectively show that tamoxifen reduces fibrosis in PDAC tissues and acts as a potential modulator of inflammatory and immune responses by directing the recruitment and polarization of tumor-associated macrophages.

### Tamoxifen inhibits spreading, cell–matrix attachment, and invasion in macrophages

To understand the effect of tamoxifen on the biomechanical and invasive properties of macrophages *in vitro*, we used RAW264.7 murine macrophage cells. The recruitment of macrophages in tissues requires macrophage–ECM attachment and further macrophage spreading. To determine whether tamoxifen affects cell spreading, we seeded macrophages on fibronectin-coated glass and monitored cell spreading for 1 h (Fig EV1A and B). We observed that the cell spreading area during this time was 30% less in tamoxifen-treated macrophages, when compared to control macrophages. Focal adhesions are complex structures that mediate communication between the cell and the ECM and are essential in cell spreading. This prompted us to study whether tamoxifen could affect focal adhesion dynamics in these cells. The number and size of the focal adhesions in tamoxifen-treated macrophages were significantly decreased compared to control macrophages (Appendix Fig S2).

Next, we seeded macrophages on fibronectin-coated polyacrylamide matrices of low (1 kPa) and high (25 kPa) rigidities[28] to better recapitulate the ECM rigidity found in healthy and PDAC tissues, and quantified cell number, cell spread area, and roundness for control and tamoxifen-treated macrophages (Fig EV1C–F). In the control group, the number of macrophages attached to the 25 kPa matrix was nearly double the number of those on the 1 kPa matrix. Furthermore, control cells on the 25 kPa matrix showed a 50% increase in cell area and a 50% decrease in roundness, compared to control cells on the 1 kPa matrix. Interestingly, tamoxifen treatment abolished the sensitivity of cell number, cell area, and roundness to matrix rigidity. For tamoxifen-treated cells on both 1 and 25 kPa matrices, values for these parameters were equivalent to control macrophages on the 1 kPa matrix.

We then sought to study whether tamoxifen affects the invasive behavior of macrophages *in vitro* using Transwell invasion assays. We observed a significant inhibition of macrophage invasion in the tamoxifen-treated group with respect to the control cells. Invasion was still inhibited when tamoxifen was used in the presence of the estrogen receptor antagonist. However, inhibition was alleviated

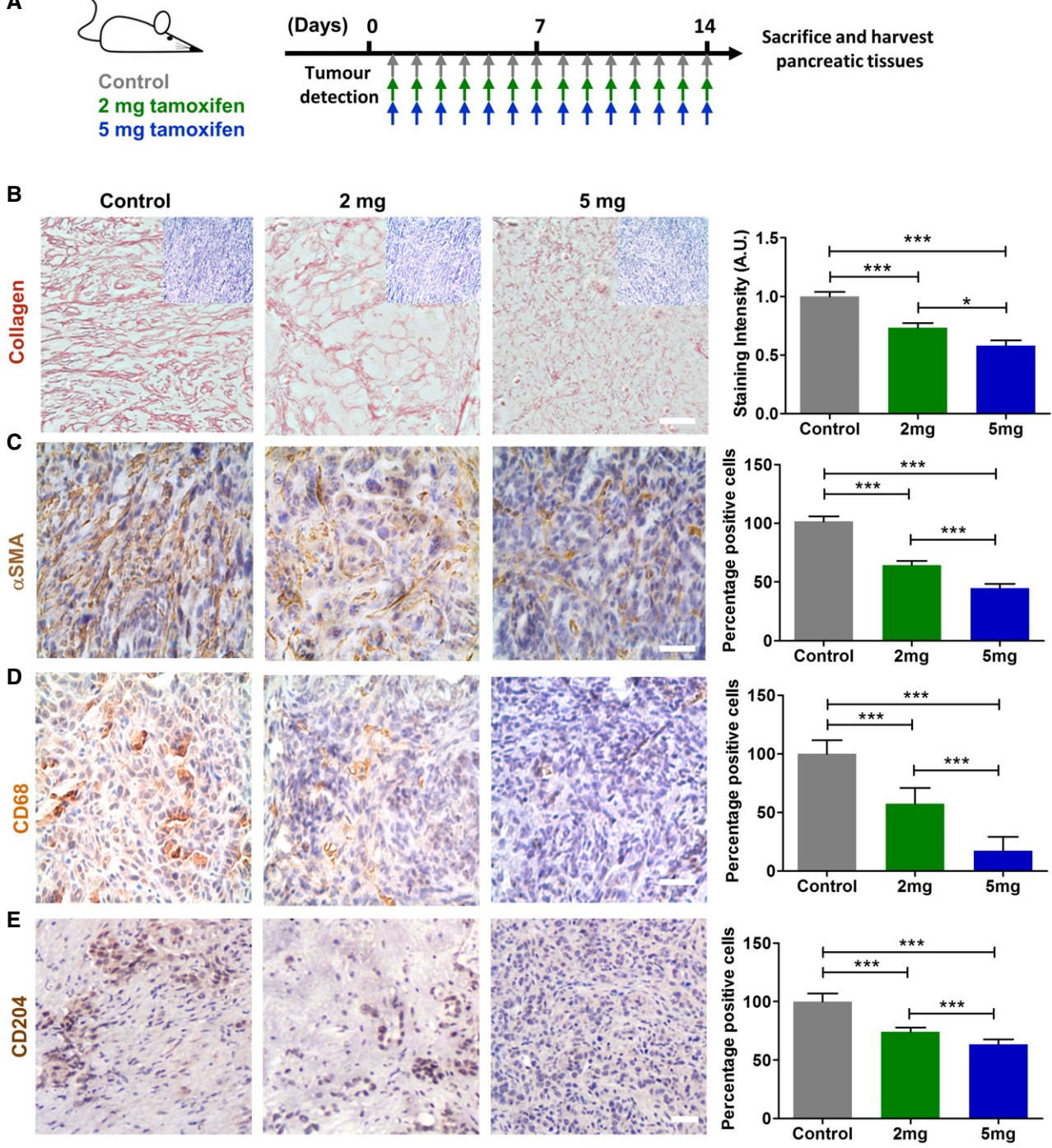

**Figure 1. Tamoxifen decreases pancreatic fibrosis and regulates macrophage infiltration and polarization.**

A Representation of mice treatment.

B–E IHC images for collagen, a-SMA, CD68, and CD204. The inserts in panel (B) represent the hematoxylin (nucleus) & eosin (cytoplasm) staining for the same tissue area of the collagen staining. For a-SMA, CD68, and CD204, the percentage of positive cells is relative to the control condition. In all cases $n$ = 4 control mice and ≥ 3 mice for 2 mg and 5 mg. Scale bars: 100 μm. The quantification for each staining is shown in the histogram in the right side. Histogram bars represent mean ± s.e.m. (ANOVA and Tukey's tests *$P$ < 0.05; ***$P$ < 0.001). For all panels, three experimental replicates.

when the GPER antagonist was used (Fig EV1G–I). This supports the notion that tamoxifen reduces macrophage invasion through GPER signaling. We also tested the effect of tamoxifen on the proliferation and apoptosis of these macrophages and observed that the proliferation rate in the treated group was twofold less than the control group (Appendix Fig S3) and that apoptosis in the treated cells occurred at double the rate observed in control cells (Appendix Fig S4). Taken together, these results show that

tamoxifen modulates focal adhesion, cell spreading, cell–ECM attachment, and GPER-mediated invasion in macrophages.

## Tamoxifen mechanically deactivates pancreatic stellate cells

To gain more insights into the molecular mechanism underpinning the tamoxifen effect in pancreatic tumor microenvironment, we focused on PSCs, which are the key effector cells of the desmoplastic reaction and display an activated myofibroblast phenotype in PDAC [29]. The persistent activation of myofibroblasts requires the establishment of a positive mechanical feedback loop, which entails the cell capacity to promote and sense a stiff environment by applying endogenous forces and mechanosensing ECM rigidity [30,31]. Annulment of this mechanical feedback loop renders PSC quiescent [10]. To determine the effect of tamoxifen on PSC activation, we studied these two properties, mechanosensing and force generation. PSCs were treated with 5 μM of tamoxifen or vehicle control for 10 days. To test the ability of PSCs to sense a mechanical external stimulus, we utilized a magnetic tweezers device to apply a pulsatile force regimen on integrin receptors of the PSCs surface using a fibronectin-coated magnetic bead (Fig 2A). Cells with an intact mechanosensing ability normally detect force application and respond to this mechanical tension by rapidly remodeling and stiffening their cytoskeleton (a process known as reinforcement) [32]. While control PSCs exhibited robust reinforcement to the applied force, as shown by a decrease in the oscillatory amplitude of the bead bound to the cell, tamoxifen-treated PSCs displayed significantly impaired reinforcement/mechanosensing (Fig 2B and C).

In addition, tamoxifen-treated PSCs were significantly softer compared to control PSCs, suggesting a decrease in overall cytoskeletal tension (Fig 2D). An increased cytoskeletal stiffness correlates with higher capacity to apply forces on their substrates [33,34]. To examine how tamoxifen affects the ability of PSCs to apply endogenous forces, we embedded PSCs in 3D collagen I/Matrigel gels and allowed them to contract the matrix. While control PSCs significantly contracted the gels after 72 h (65% gel contraction), gels with tamoxifen-treated PSCs showed a severely reduced contraction (35% gel contraction), thereby confirming that tamoxifen inhibits the ability of PSCs to apply forces and contract the matrix. The difference in the ability of control and tamoxifen-treated PSCs to deform the gels was not observed when PSCs were treated with Y-27632, a potent inhibitor of calcium-independent Rho-dependent acto-myosin contraction [35] (Fig 2E). Interestingly, tamoxifen also affected the ability of hepatic stellate cells (HSCs) and human

foreskin fibroblasts (HFFs) to deform the matrix (Fig 2E), which, together with previous work [13], argues for a broader role of tamoxifen as a regulator of cell mechanics.

We further investigated PSC activation after tamoxifen treatment. α-SMA and vimentin, two widely used markers for myofibroblast activation, were markedly reduced in tamoxifen-treated PSCs. In contrast, desmin, a marker of PSC quiescence [36], was more highly expressed (Appendix Fig S5). Collectively, these data show that tamoxifen impairs the ability of PSCs to sense external mechanical stimuli and consequently ECM rigidity, decreases overall cytoskeletal tension by increasing cell elasticity, and decreases the capacity of PSCs to apply contractile forces and deform the underlying matrix in an acto-myosin-dependent manner.

## GPER mediates tamoxifen biomechanical deactivation of PSCs

Tamoxifen is an estrogen analogue and as such is widely known to mediate its physiological effect through the classical estrogen receptors alpha and beta (ERα and ERβ), which translocate to the nucleus once activated and act as ligand-dependent transcription factors [37]. However, GPER (G protein-coupled estrogen receptor) from the GPCR family of receptors has been recently shown to modulate the non-genomic cell responses to estrogen in a broad variety of cell types [11,38–41]. We used immunofluorescence and immunoblotting to confirm that PSCs express ERα, ERβ, and GPER (Appendix Fig S6). To determine which of these specific receptors mediate the biomechanical deactivation of PSCs by tamoxifen, we pre-treated PSCs with ICI182780 (the commonly used ER antagonist); or G15 (a selective antagonist for GPER), prior to and during tamoxifen treatment, and tested the two properties that promotes the positive feedback loop, which activates PSCs: mechanosensing (using the magnetic tweezers protocol previously described) and force generation (using the 3D gel contraction assay). We observed that in the presence of the ER antagonist, tamoxifen can still impair mechanosensing (Fig 2F). However, this effect is abrogated in the presence of the GPER antagonist (Fig 2G). Similarly, tamoxifen severely reduced PSC-mediated gel contraction when the ER antagonist, but not GPER antagonist, was present (Fig 2H). Furthermore, we observed that treating PSCs with PPT or ERB041, two selective agonists for ER-α and ER-β, respectively, did not affect PSC morphology or the ability to contract 3D matrices. However, G1, a potent selective agonist for GPER led to profound changes in morphology; PSCs adopted a much less elongated-contractile phenotype and exhibited a significantly reduced capacity to contract 3D

---

**Figure 2.  Tamoxifen impairs mechanosensing and force generation via GPER.**

A       Representation of the magnetic tweezers.
B       Representative traces tracking bead displacements.
C       Histogram shows relative bead displacement for the first and last pulse, *n* = 27 control and *n* = 37 tamoxifen.
D       Quantification of PSCs stiffness, *n* = 40 control and 38 tam cells.
E       Histogram shows percentage of gel contraction, *n* > 10 gels per condition.
F, G    Representative mechanosensing trace of PSC treated with tamoxifen and estrogen receptor (ER) or G protein-coupled estrogen receptor (GPER) antagonists. Histogram shows relative bead displacement for the first and last pulse, *n* > 20 for all conditions.
H       Histogram shows percentage of gel contraction by PSCs treated with tamoxifen in the presence of indicated antagonist, *n* > 10 gels per condition.
I       Histogram shows percentage of gel contraction after treatment with indicated agonist, *n* > 10 gels per condition.

Data information: (E, H, I) Representative images above, dotted yellow lines represent the gel contour. (B, F, G) For mechanosensing: black/red arrows indicate initial and final amplitude of the bead oscillation. In all cases, histogram bars represent mean ± s.e.m. (ANOVA and Tukey's test in H and I and *t*-test for the rest *P* < 0.05; **P* < 0.01; ***P* < 0.001). Three or more experimental replicates in all cases.

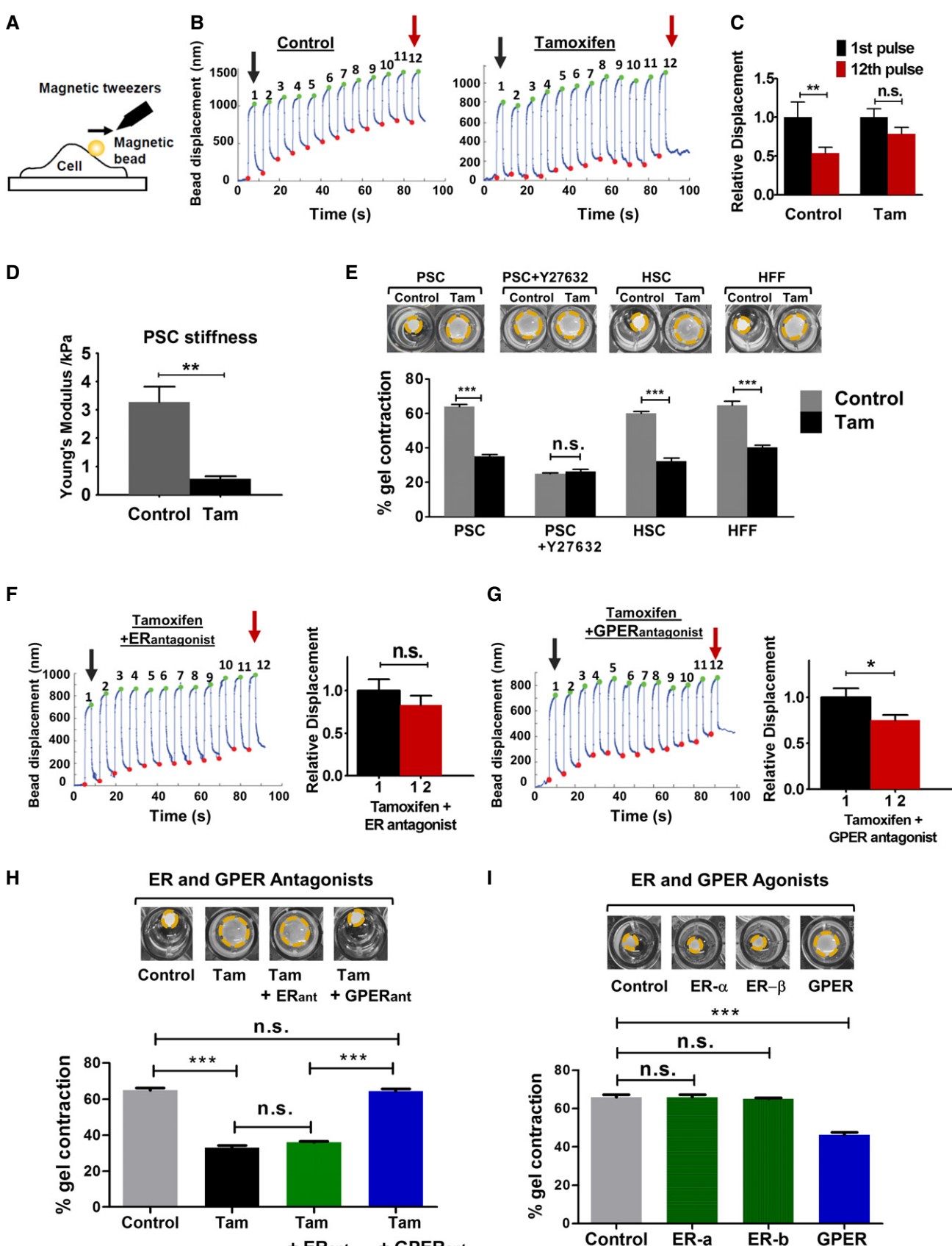

Figure 2.

matrices (Figs 2I and EV2). We also found that blocking GPER, but not ERs, suppresses the previously observed effect of tamoxifen on the phosphorylation and activation of MLC-2 (Appendix Fig S7).

## Tamoxifen inhibits ECM remodeling by PSCs and cancer cell invasion

The ability of fibroblasts to contract collagen gels correlates with their ECM remodeling capacity [42], and stiff ECM promotes breast cancer invasion via enhanced integrin-mediated mechanosignaling [43]. To examine the effect of GPER on ECM remodeling by PSCs, we treated PSCs with tamoxifen, embedded PSCs in 3D collagen I/ Matrigel matrices, allowed them to remodel the matrix for 72 h, and utilized AFM to measure the elastic modulus of the remodeled matrices. While remodeling by control PSCs increased the stiffness of the ECM fivefold, tamoxifen treatment markedly decreased the capacity of PSCs to stiffen the ECM (Fig 3A). Organotypic models were used to investigate the effect of impaired ECM mechanical remodeling on promoting pancreatic cancer cell invasion. To do this, we used 3D matrices previously remodeled by PSCs as described earlier in this section. After remodeling, PSCs were killed, removed from the matrix, and AsPC1 pancreatic cancer cells were seeded on top of the gels and allowed to invade the mechanically remodeled matrices (Fig 3B and C, and Appendix Fig S8). Pancreatic cancer cells invaded deeply into the matrix previously remodeled by control PSCs, but showed a minimal capacity to invade matrices remodeled by tamoxifen-treated PSCs. These data suggest a potential use of tamoxifen and GPER agonists to restore tissue homeostasis to pancreatic tissues by modulating ECM remodeling by PSCs and to abolish cancer cell invasion.

## Tamoxifen deactivates YAP in PSCs and pancreatic tissues

Increased matrix stiffness promotes the activation of Yes-associated protein (YAP), which is a key mechanotransducer required for myofibroblast induced matrix remodeling and cancer cell invasion in breast cancer [42,44]. YAP activation, a hallmark of activated myofibroblasts, leads to YAP nuclear localization, while the inactive form (phosphorylated on residue Ser127) is sequestered in the cytoplasm [44]. We used immunofluorescence to quantify the levels of active and inactive YAP in control and tamoxifen-treated PSCs. Tamoxifen treatment reduced active nuclear YAP in PSCs by more than half (Fig 4A and B). We also observed a substantial reduction in the activation of YAP target genes CTGF and ANRKD1 (Fig 4C). We used immunoblotting to quantify the total levels of YAP and its inactive form pYAP Ser127 in the cytoplasm (Fig 4D–E and Appendix Fig S9). Both total YAP and pYAP Ser127 were reduced in PSCs in response to tamoxifen by approximately 33% and 22%, respectively. However, pYAP Ser127 was reduced to a lesser extent such that the ratio between pYAP and total YAP actually increased by approximately 17% in the tamoxifen-treated PSCs. Furthermore, YAP is abundantly expressed in stellate cells of both human PDAC precursor lesions (PanIN) and PDAC tissues and in Kras G12D mutant mice tissues [45]. Importantly, YAP activation has been recently shown to trigger Kras-independent PDAC maintenance [46], and in Kras-dependent PDAC, YAP is critically required for progression to invasive PDAC in mice [47]. Thus, we used immunohistochemistry (IHC) to quantify the levels of YAP active nuclei in tissues from KPC mice treated with tamoxifen. While 75% of nuclei were active for YAP in tissues from control KPC mice, only 65% and 38% nuclei were active for YAP in tissues from KPC mice treated with 2 and 5 mg of tamoxifen, respectively (Fig 4F and G). To further validate these results, we used immunofluorescence to double stain YAP with αSMA (as a PSC specific marker). There was a 50% decrease in the nuclear versus cytoplasmic YAP in the tissues coming from mice treated with tamoxifen with respect to tissues from control mice (Appendix Fig S10). These data show that tamoxifen significantly reduces the levels of active YAP and its localization in PSC nuclei *in vitro* and in mouse models of PDAC.

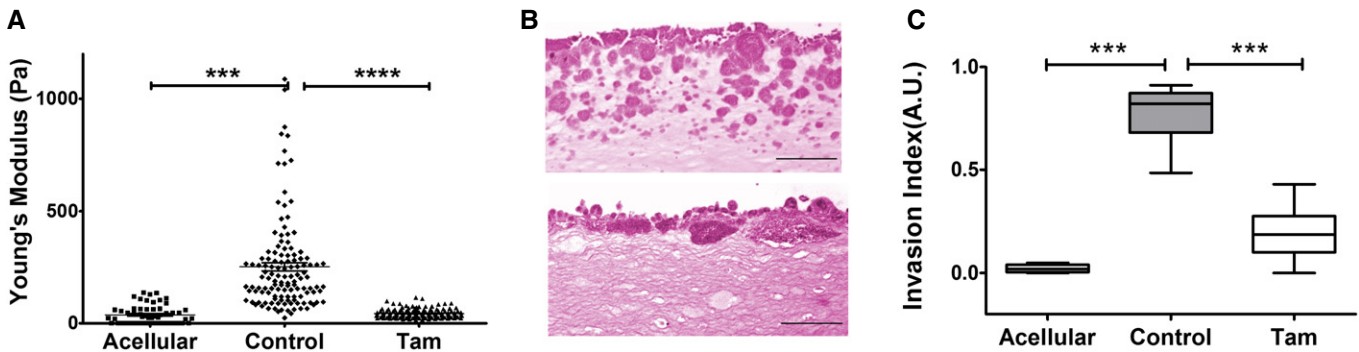

**Figure 3. Tamoxifen suppresses ECM remodeling by PSCs and cancer cell invasion.**

A   Young's modulus for matrices previously remodeled by PSCs; control *n* = 126 and tamoxifen *n* = 145. Lines and error bars indicate mean ± s.e.m. Four experimental replicates.

B   Representative images of H&E staining showing AsPC1 cancer cell invasions in matrices previously remodeled by control PSCs (top) or tamoxifen-treated PSCs (bottom), scale bars 50 μm.

C   AsPC1 invasion quantification, the central box represents values from the lower to upper quartile. The middle line represents the mean. The vertical line extends from the minimum to the maximum value, *n* = 30 control and *n* = 40 tamoxifen.

Data information: ANOVA and Tukey's test, ***P < 0.001; ****P < 0.0001. Three experimental replicates for (B, C).

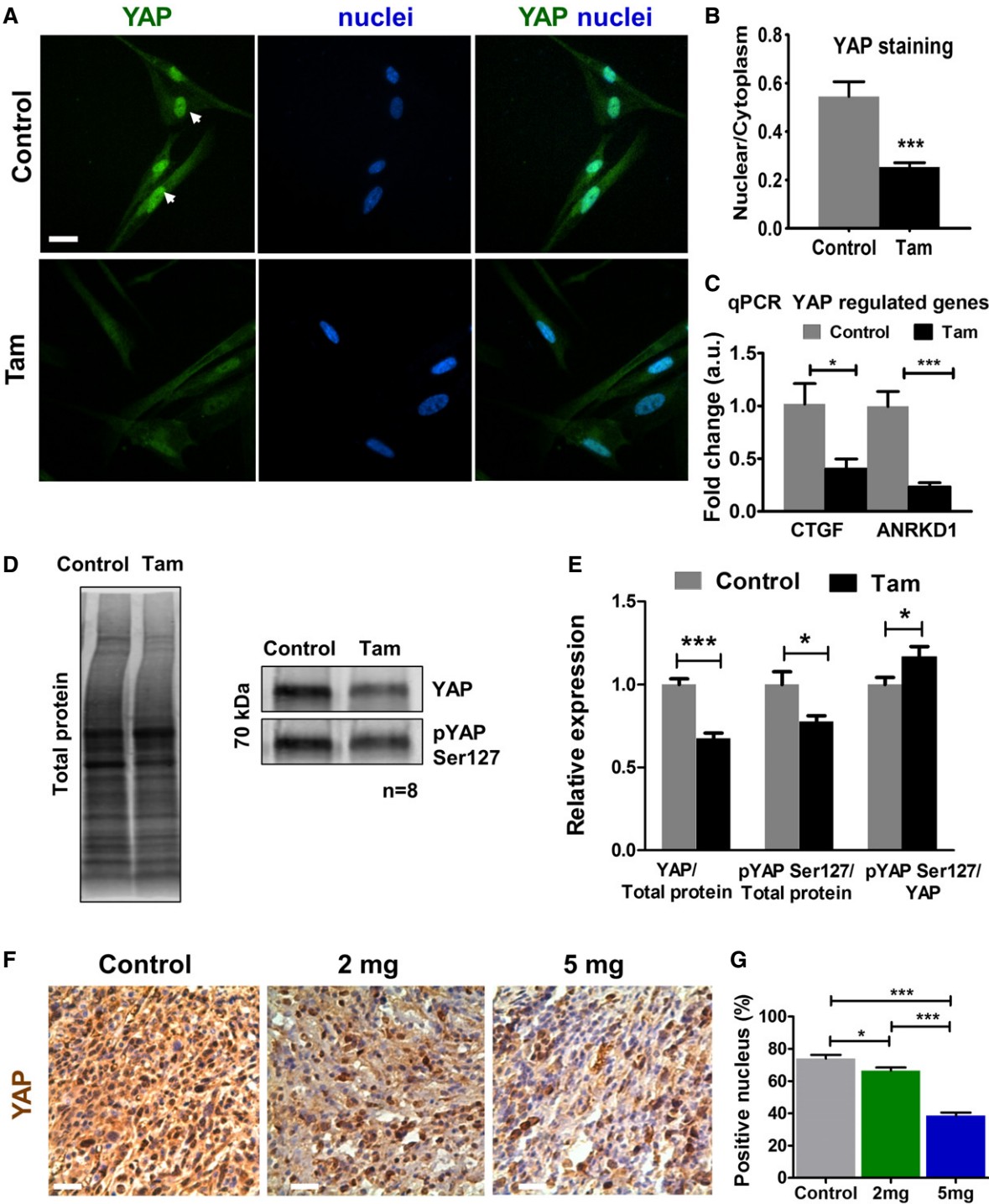

**Figure 4. Tamoxifen deactivates YAP in PSCs and in pancreatic tissues.**

A   Immunofluorescence images of PSCs stained for YAP. The white arrows show YAP localization in the nucleus. Scale bar: 20 μm.
B   Quantification of the nuclear/cytoplasm YAP in PSCs (four experimental replicates).
C   qPCR mRNA levels for YAP target genes connective tissue grow factor (CTGF) and ankyrin repeat domain 1 (ANRKD1) (three experimental replicates).
D   Western blot bands for YAP, pS127 YAP, and total protein.
E   Quantification of YAP and pYAP Ser127 normalized to total protein, expressed relative to unstimulated control (*n* = 8).
F   IHC images for YAP staining in KPC mice pancreatic tissues (control and treated with 2 and 5 mg tamoxifen), YAP staining is brown. Scale bars: 100 μm.
G   Quantification of YAP-positive nucleus. *n* = 5 control mice, and ≥ 3 mice for 2 mg and 5 mg.

Data information: In all cases, bars indicate mean ± s.e.m. *t*-test for (B, C, and E) and ANOVA and Tukey's test for (G), *\*P* < 0.05; *\*\*\*P* < 0.001. Three experimental repeats.

## GPER activation decreases MLC-2 activation and PSC migration

We then sought to elucidate the mechanism by which GPER targets the capacity of cells to respond to external mechanical stimuli and to apply endogenous forces to the ECM using the GPER agonist tamoxifen. Both properties critically depend on the cell's contractile acto-myosin machinery [32,48]. We used immunofluorescence and Western blot to confirm that while the total levels of myosin light chain 2 (MLC-2) remained unaltered in tamoxifen-treated PSCs with respect to control PSCs, the levels of phosphorylated MLC-2 (active form) were significantly decreased in tamoxifen-treated PSCs (Fig 5A–C and Appendix Fig S11). We utilized immunoassays to quantify the levels of total and activated RhoA, which activates Rho-associated kinase (ROCK) to promote MLC-2 phosphorylation and activation (Fig 5D). The total levels of RhoA did not change significantly between control and tamoxifen-treated PSCs, but the active levels of RhoA (GTP bound RhoA) were decreased (twofold) in tamoxifen-treated PSCs in comparison with control PSCs. We therefore expressed constitutively active RhoA in tamoxifen-treated PSCs, which rescued gel contraction to the same extent as control PSCs (Appendix Fig S12). To validate these findings in pre-clinical models, we quantified the levels of active phosphorylated MLC-2 in stromal pancreatic tissues from KPC mice treated with tamoxifen. We observed a marked and significant tamoxifen dose-dependent decrease in pMLC-2 levels in these tissues (Fig 5E and F). We further used double staining of pMLC-2 and αSMA (PSC marker) and found a similar pronounced 70% reduction of pMLC-2 in PDAC tissues coming from tamoxifen-treated mice (Appendix Fig S13).

Additionally, stellate cells infiltrate the stroma and migrate toward the tumoral area to cross-talk with cancer cells and promote tumor progression [49,50]. Cancer-associated fibroblasts promote the formation of metastatic niches in distant organs [51] by generating tracks in the ECM that facilitate tumor epithelial cells colonization. We therefore investigated the effect of tamoxifen treatment on PSC migration and observed that tamoxifen severely decreased (sixfold) PSC migration (Appendix Fig S14). Thus, by preventing PSC migration tamoxifen may hamper the formation of metastatic niches and annulling the communication between PSCs and cancer cells.

Finally, we used siRNA against GPER to knock down its expression in PSCs and observed that tamoxifen treatment on the GPER knock down PSCs cannot decrease the levels of active myosin (pMLC-2) and does not inhibit myofibroblast differentiation in PSCs

(measured by the PCS activation markers αSMA and vimentin) as it does in wild-type PSCs (Appendix Fig S15). Taken together, these data show that tamoxifen restores non-fibroblastic morphology in PSCs via canonical RhoA-mediated down regulation of MLC-2 activation.

Although the desmoplastic stroma has long been considered an attractive target for PDAC therapy, its overall contribution to the disease progression is far from settled and radical approaches that completely ablate the stroma do not seem to be an effective option [5,9]. Instead, targeted stromal modulation that restores mechanical homeostasis may hold significant therapeutic value.

In this regard, our results show that tamoxifen treatment abolishes the biomechanical feedback loop that sustains PSC activation via GPER signaling, and by deactivating PSCs, PSC-mediated ECM remodeling and cancer cell invasion are impaired. Interestingly, in our study cancer cell invasion was assessed in matrices, which were deprived of both the remodeling PSCs and any soluble factor that might allow PSC–cancer cell communication. This suggests that under tamoxifen treatment, PSCs are unable to biomechanically and topologically remodel the ECM, and the resulting ECM architecture is not as conducive to cancer cell invasion. This idea is consistent with previous works that have shown that increased ECM stiffness can promote cancer cell invasion [43] and that RhoA-mediated fibroblast ECM remodeling enhances invasion [51,52]. Also, increased alignment and thickness of the collagen fibers enhance cancer cell invasion [53]. Tamoxifen also reduces the expression of the cross-linking enzyme LOX-L2 and the matrix metalloproteinase MMP-2 in PSCs [54], which suggests that it is not only the ECM rigidity what influences cancer cell invasion, but that other topological factors play an important role, such as the presence of tracks promoted by MMPs [51] and the alignment and thickness of collagen fibers [53].

Tamoxifen reduces tissue stiffness, collagen deposition, collagen fiber thickness, and αSMA expression in PDAC mouse models. Concurrently, tamoxifen impedes the recruitment of tumor-associated macrophages and their polarization toward the M2 phenotype that are highly associated with invasion and metastasis in PDAC [22,26]. A summary of these interactions is shown in Fig 5G. We acknowledge that our *in vivo* studies focused on high-dose tamoxifen administration, and scaling this dose based on body weight in humans would result in supraphysiologic doses, for which limited safety data exit. Therefore, future studies using lower doses are required for further clinical validation.

---

**Figure 5.  Tamoxifen decreases phosphorylation of mlc-2 in PSCs and mice pancreatic tissues and suppresses RhoA activation in PSCs.**

A   IF images for total (MLC-2) and phosphorylated (pMLC-2) in PSCs, scale bar is 20 μm.
B   Quantification of staining in (A). Each data point represents a cell.
C   Western blot for total and phosphorylated mlc-2.
D   Quantification of total and active RhoA, expressed as percentage of the total RhoA in the control condition.
E   IHC images for phosphorylated mlc-2 in mice pancreatic tissues (*n* = 5 control, *n* = 5, 2 mg, and *n* = 3, 5 mg). Scale bars: 100 μm.
F   Quantification of staining in (E).
G   Illustration of the effect of tamoxifen on GPER activation at the cellular level (left panel) and tissue level (right panel). In PSCs, tamoxifen suppresses the activation of YAP and MLC-2 (by phosphorylation to pMLC-2) via the axis GPER/RhoA. This inhibits mechanosensing and the ability to apply endogenous forces in PSCs, which are required to maintain the myofibroblastic phenotype in PSCs. Consequently, PSCs adopt a mechanically inactive state (not myofibroblast-like cell). Pancreatic tissues from KPC mice treated with tamoxifen have reduced tissue stiffness and desmoplastic reaction (decreased collagen deposition). The expression of the myofibroblast marker a-SMA is reduced consistent with the mechanical inactivation of PSCs. The recruitment of macrophages, their M2 polarization, and the invasion of cancer cells are also reduced in these tissues.

Data information: In all cases, histogram bars indicate mean ± s.e.m. *t*-test for (B, C, and D) and ANOVA and Tukey's test for (F), *$P < 0.05$; ***$P < 0.001$. Three or more experimental repeats for all panels.

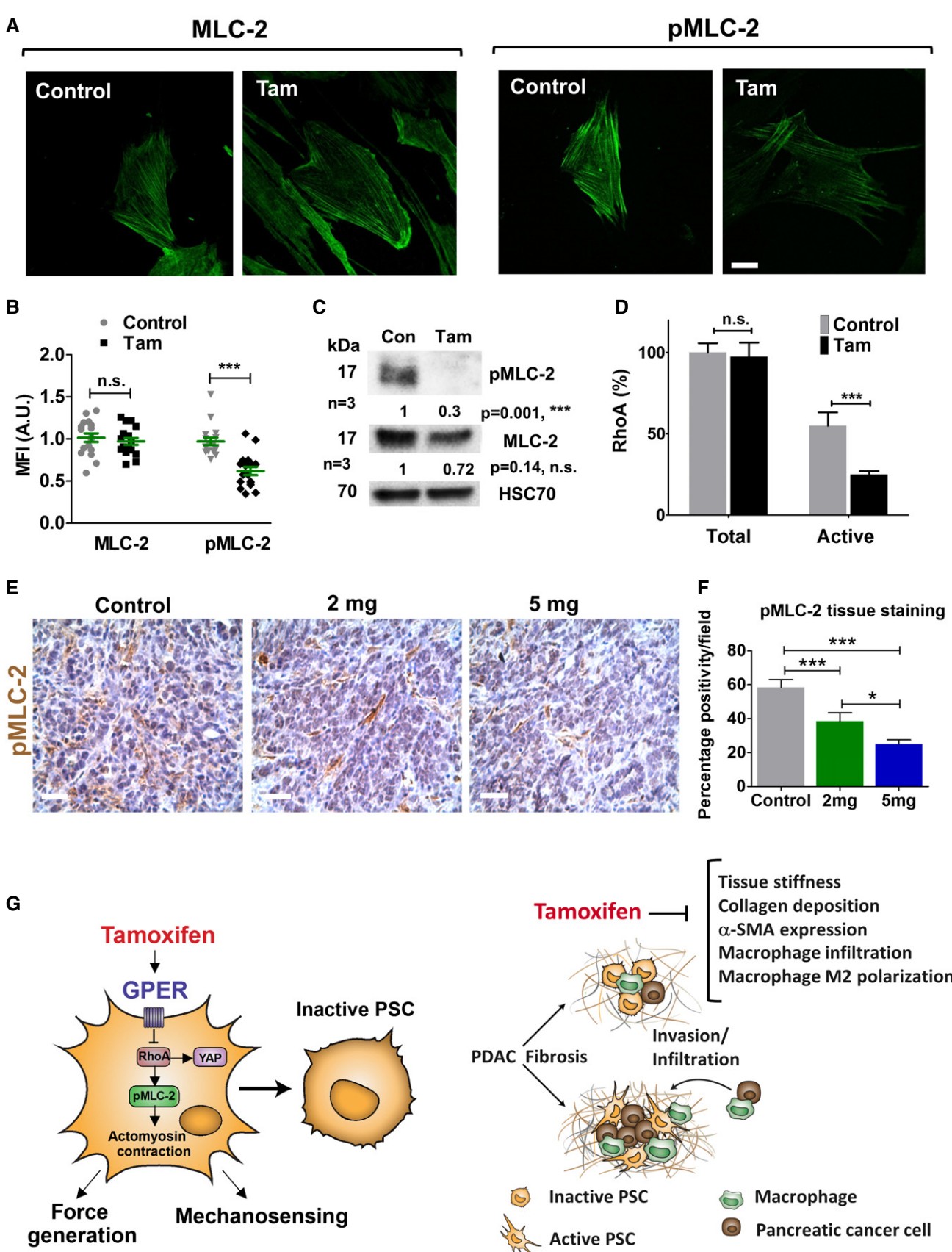

**Figure 5.**

Most solid carcinomas, such as PDAC, are linked to developed fibrosis, which is driven by myofibroblast-like cells in the tumor microenvironment. To be able to sustain fibrosis, these cells develop a robust contractile phenotype that requires the activation of MLC-2 [1,55]. The reported effects of GPER on cell mechanics targeting key molecules in cellular mechanotransduction such as RhoA, MLC-2, and YAP highlight the potential of this receptor as an effective mechanoregulator of the tumor microenvironment. Considering that GPER is broadly expressed across tissues, the pleiotropic effect of estrogens, the commonalities of GPCR signaling, and the proven safety of tamoxifen in the clinic, it is possible that tamoxifen may lead a new stromal reprogramming strategy to target the myofibroblast-like cells in the tumor microenvironment. Certainly, an increased appreciation of GPER as a convergence point for multiple environmental factors in the tumor microenvironment is expected in the coming years.

# Materials and Methods

### Mice

KPC mice (Pdx-1 Cre, Kras$^{G12D/+}$, p53$^{R172H/+}$) were randomized to three groups and were injected (IP) with either (i) vehicle [corn oil], (ii) 2 mg, or (iii) 5 mg of tamoxifen daily starting the same day when PDAC tumor was detected and continuing until mice reached endpoint (for most mice between 8–14 days). Control: 3 males/2 females; 2 mg: 3 males/3 females; and 5 mg: 2 males/2 females. After the treatment, mice were sacrificed and pancreatic tissues harvested and used for further analysis. Animals were maintained in conventional animal facilities and monitored daily. All studies were conducted in compliance with the UK Home Office guidelines under license and approved by the local ethical review committee.

### Cell culture and antibodies

Primary, culture-activated human pancreatic stellate cells (passages 6–8, HPaSteC #3830-Caltag Medsystems, UK) were exposed to tamoxifen (Tam) dissolved in ethanol at a concentration of 5 µM for 10 days. Medium was changed every 72 h and the drug treatment was performed in subdued light. Tamoxifen (Z-4-hydroxytamoxifen) was from Sigma-Aldrich (H7904). Cells incubated with culture medium (Dulbecco's modified Eagle's medium—DMEM with 2% FBS, 1% penicillin/streptomycin, and 1% fungizone antimycotic) with an equivalent amount of vehicle (0.1% ethanol) served as controls. AsPC-1 cells were grown in RPMI 1640 with 10% fetal bovine serum heat inactivated (Gibco, Loughborough, UK), 2 mM L-glutamine, 1% penicillin/streptomycin (Sigma-Aldrich), and 1% fungizone amphotericin B (Gibco, UK). RAW264.7 murine macrophages were maintained in DMEM with 4.5 g/l glucose, L-glutamine, 10% fetal bovine serum (FBS; Sigma-Aldrich), and 1% penicillin/ streptomycin. All cells were tested for mycoplasma contamination. Antibodies: HSC70—Santa Cruz, MLC-2—Millipore MABT180, p-MLC-2 (Thr18/Ser19)—Cell Signaling, 3674, YAP—Santa Cruz sc-101199, GPER antibody—Abcam ab39742, paxillin—BD Biosciences 610051, anti-mouse HRP—Invitrogen 626580, anti-rabbit HRP— Abcam ab137914, anti-mouse 488, Invitrogen—A11029. Agonists and antagonists: All were purchased from Tocris and used at 1 µM,

ICI182780 (cat. 1047), G15 (cat. 3678), G1 (cat. 2577), PPT (cat. 1426), ERB041 (cat. 4276), siRNA for GPER (Santa Cruz Biotechnology, cat. sc-60743). For the experiments using the AsPC1 conditioning media, cells were grown under normal culture media with 10% FBS until 80% confluency, washed 3 times with PBS, and grown in serum free media for 24 h. The medium was collected and used to grow PSCs for 24 h in media with 40% conditioned media and 60% PSC media.

### Atomic force microscopy

For AFM study, collagen Matrigels were lifted from the 96-well plates prior to measurement and immediately attached to a Petri dish with a droplet of cyanoacrylate adhesive, applied with a 10-µl pipette tip. After Matrigel attachment (1–2 min), the slice was immersed in culture medium (DMEM with 2% FBS) in order for the AFM measurements to be conducted within a 2-h time period. Measurements of the Matrigels have been conducted on a JPK NanoWizard-1 (JPK Instruments) operating in force spectroscopy mode, mounted on an inverted optical microscope (IX-81; Olympus). AFM pyramidal cantilevers (MLCT; Bruker) with a spring constant of 0.07 N/m were used with a 35-µm glass bead attached to cantilever tip. Prior to measurements with the adapted cantilevers, their sensitivity was calculated by measuring the slope of force–distance curve in the AFM software on an empty region of the Petri dish. For indentation tests on the sample, the cantilever was aligned over regions in the middle of the samples using the optical microscope. For each sample, 30 force curves were acquired across 6 different 100-µm regions, and this arrangement allowed force curves to be acquired in locations at least 50–100 µm apart. Force-curve acquisition was carried out with an approach speed of 5 µm/s and a maximum set force of 1.5 nN. Elastic moduli were calculated from the force–distance curves by fitting the contact region of the approach curve with the Hertz contact model, using the AFM software. For AFM study on animal tissues, small animal tissue cubes were attached to the Petri dish in a similar way to Matrigels. A small amount of cyanoacrylate adhesive was applied with a 10-µl pipette tip to which the tissue was applied and left to adhere for 1–2 min. AFM on tissue was conducted in PBS with the same force spectroscopy settings as Matrigels.

### Cell mechanosensing

To assess how PSCs (control or Tam-treated) sense and respond to applied forces emanating from the ECM, 4.5-µm fibronectin-coated magnetic beads coated were subjected to a pulsatile force regimen applied with magnetic tweezers, consisting of a 3-s, 1 nN pulse of force, followed by a 4-s period of rest, repeated for 12 total pulses for 100 s. The ability of the cells to sense and respond to the applied tension was examined from the rapid cell stiffening response evident by the progressive decrease in amplitude of the bead movement ($n = 27$ for PSC Control and $n = 37$ Tam).

### Cell compliance

To characterize the mechanical properties of PSCs, we used magnetic tweezers microrheology to measure cell deformation in response to magnetically generated forces. Superparamagnetic 4.5 µm epoxylated beads (Dynabeads, Life Technologies) were coated with fibronectin (40 µg per $8 \times 10^7$ beads) and incubated

with adherent cells for 30 min, prior to measurements, to allow integrin binding and provide a mechanical link between the bead and the cytoskeleton. The unbound beads were removed by multiple washing with PBS. The experiments were performed at 37°C, 5% $CO_2$, and 95% humidity in DMEM containing 2% FBS in a microscope stage incubation chamber. A viscoelastic creep experiment was conducted by applying mechanical tension onto single beads bound on the apical surface of the cells with a constant pulling force ($F_0$ = 3nN) for 3 s generated by the magnetic tweezers. The viscoelastic creep response of cells was recorded by tracking the resulting bead displacement in bright field (40× objective at 20 frames per second) that is indicative of the local cell deformation. The viscoelastic creep response J(t) of cells during force application followed a power law in time $J(t) = J_0(t/t_0)^\beta$ with the prefactor $J_0$ representing cell compliance ($J_0$ = inverse of cell stiffness in units of $kPa^{-1}$). The creep compliance J(t) of the cell is essentially the ratio ($\gamma(t)/\sigma_0$) of the localized cellular strain $\gamma(t)$ induced by the applied stress from the magnetic tweezers $\sigma_0$, with $\gamma(t)$ taken as the radial bead displacement normalized over the bead radius $\gamma(t) = d(t)/r$ and the applied stress as $\sigma_0 = F_0/4\pi r^2$ taken as the applied force normalized over the bead cross-sectional area.

## Real-time quantitative polymerase chain reaction

Total RNA was extracted with RNeasy Mini Kit (Qiagen, 74104), and 1 μg of total RNA was reverse-transcribed by High-Capacity RNA-to-cDNA Kit (Applied Biosystems, 4387406) according to manufacturer's instructions. qPCR was performed with SYBR Green PCR Master Mix (Applied Biosystems, 4309155) with 100 ng cDNA input in 20 μl reaction volume. GAPDH expression level was used for normalization as a housekeeping gene. The sequences were as following: GAPDH: forward-5′ACAGTTGCCATGTAGACC-3′, reverse-5′TTTTTGGTTGAGCACAGG-3′; a-SMA: forward-5′CATCA TGAAGTGTGACATCG-3′, reverse-5′GATCTTGATCTTCATGGTGC-3′; CTGF: forward-5′-TTAAGAAGGGCAAAAAGTGC-3′, reverse-5′-CATACTCCACAGAATTTAGCTC-3′; ANKDR1: forward, 5′-TGAGTA TAAACGGACAGCTC-3′ and reverse, 5′-TATCACGGAATTCGAT CTGG-3′. All primers were used at 300 nM final concentration. The relative gene expression was analyzed by comparative $2^{-\Delta\Delta C_t}$ method.

## Immunofluorescence staining

All immunofluorescence staining was done on coverslips coated with 10 μg/ml fibronectin (Sigma, F0895). Following pertinent treatment, cells were fixed with 4% PFA (Sigma, P6148) in PBS for 10 min then blocked and permeabilized with 2% BSA-0.1% Triton (Sigma, T8787) in PBS for 30 min. After blocking, cells were incubated with primary antibodies prepared in blocking solution for 1 h at room temperature in a humidified chamber. Then, cells were washed in PBS and incubated with Alexa Fluor 488-conjugated secondary antibodies and Phalloidin (Invitrogen, A22283, 1/1,000 dilution) prepared in PBS for 30 min at room temperature. Finally, coverslips were washed in PBS and mounted in mounting reagent with DAPI (Invitrogen, P36931). For tissue staining, FFPE blocks were first sectioned, deparaffinized, and rehydrated as described previously. Antigen retrieval was done by incubating the sections in boiled citrate buffer pH = 6 for 1 h.

## YAP IF measurements

Yes-associated protein immunofluorescence studies were conducted on four different samples incorporating more than 8 separate regions of interest to obtain images of single cells. IF images were obtained for PSCs in the two populations stained with Alexa fluor 488, using a fluorescence microscope (AE31 trinocular, Motic) with a 480/30-nm excitation filter and a 535/40-nm emission filter. Images were obtained with a CMOS camera (Moticam 5, Motic) for 40 regions across the different coverslips with each population. Images for DAPI were also obtained with an excitation filter of 350/50 nm and emission of 460/50 nm in order to visualize the nucleus for the quantification of YAP staining regions. Images for the YAP and DAPI channels were combined to allow accurate location of the nucleus for the analysis of images in ImageJ. Measurements of the intensity of the fluorescence in the nucleus were obtained in ImageJ and compared against the total cell fluorescence intensity with the nuclear staining removed. For tissue staining: Images of the nuclei stained with DAPI were thresholded, resulting in a binary mask of the nuclei. This mask was overlaid on the YAP immunofluorescence image and the mean fluorescent intensity was measured giving the average YAP expression in the nucleus. Subsequently, the binary mask was created from the YAP immunofluorescence staining. To obtain the cytoplasmic YAP expression, the nuclear mask was subtracted from the YAP mask. The ratio between the first and the second value gave the nuclear to cytoplasmic YAP expression ratio. The values obtained from the different regions of the same tissue sections were averaged and treated as an experimental replicate. The final results were calculated for $n$ = 3 animals per condition.

## Immunohistochemistry

Formalin-fixed paraffin-embedded (FFPE) blocks were sectioned at 4 μm. Deparaffined and rehydrated with histoclear (National diagnostics, HS-200) followed by decreased concentrations of ethanol, heat-induced antigen retrieval was done by boiling the sections in 10 mM citrate buffer, pH = 6, for 20 min in microwave (for CD68 sodium citrate buffer, pH = 6, was used). Endogenous hydrogen peroxide activity was quenched with 0.3% $H_2O_2$ in methanol for 30 min at room temperature. Sections were blocked with normal serum for 1 h at room temperature and endogenous biotin activity and avidin binding sites were blocked with avidin–biotin blocking kit according to manufacturer's instructions (Vector laboratories, SP-2001). Sections were then incubated with primary antibodies diluted in blocking serum overnight at 4°C in a humidified chamber. Primary antibody dilutions are as follows: pMLC-2 (Cell Signaling, 3671, 1/200), α-SMA (Abcam, ab5694, 1/300), CD68 (AbD Serotec, MCA1957GA, 1/500), YAP (Santa Cruz sc-101199, 1/100). Primary antibodies were washed with PBS, and biotinylated secondary antibodies were diluted 1/250 in PBS and incubated for 30 min at room temperature then washed. Sections were incubated with avidin layer (VECTASTAIN Elite ABC Kit, Vector laboratories, PK-6100) for 30 min at room temperature then developed with peroxidase substrate (DAB substrate Kit, Vector laboratories, SK-4100). Finally, sections were counter-stained with hematoxylin (Abcam, ab128990), dehydrated by increasing concentration of ethanol and histoclear, and then mounted in DPX mounting (Sigma, 06522). For CD204 staining (AbD Serotec, MCA1322T, 1/250), freshly frozen

pancreas tissue was cut at 6μm and fixed with ice-cold acetone (Sigma, 650501) for 15 min at room temperature and blocked and stained as explained above. For Sirius Red staining, sections were deparaffinized and rehydrated as explained then stained with 0.1% Sirius Red (Direct Red 80, Sigma, 365548) prepared in aqueous picric acid (Sigma, P6744) for 1 h at room temperature. After washing in 0.5% acetic acid (Sigma, A6283) in water (acidified water), sections were dehydrated and mounted as explained.

Whenever possible images were taken from the stroma following the criterion explained in Appendix Fig S16. For pMLC-2, α-SMA, CD68, and CD204 staining images were divided into four equal regions of interest (ROI) and positively stained cells were counted for each ROI. For YAP staining images were divided in ROIs as explained and cells were grouped in nuclear, cytoplasmic, and negative in each ROI. Staining intensities were measured with color deconvolution. In order to analyze collagen fibril thickness and alignment in Sirius Red staining; the images were thresholded then converted into binary. Fibril thickness and alignment were analyzed on their corresponding maps, which were created with BoneJ plugin by Fiji.

## Western blotting

The cell lysates were prepared with radio immunoprecipitation assay (RIPA) buffer (Sigma, R0278) containing proteinase inhibitors (Sigma, P4340). The protein concentration was quantified by DC protein assay (Bio-Rad, 500-0113) according to manufacturer's instructions. Samples were separated by an SDS–PAGE gel under reducing conditions and transferred to a nitrocellulose membrane (GE Healthcare, 10401196) then blocked with 5% bovine serum albumin (BSA, Sigma, A8022)—0.1% Tween-20 (Sigma, P1379) in PBS. All primary antibodies were prepared in blocking solution and incubated overnight at 4°C. The membrane was washed and incubated with horseradish peroxidase (HRP)-conjugated secondary antibodies in blocking solution for 1 h at room temperature. Finally, the membrane was washed and developed with HRP substrate (Millipore, WBLUR0100).

## Invasion assays

To assess the effect of tamoxifen treatment on PSC-driven ECM remodeling related cancer cell invasion 3D organotypic cultures were employed. Organotypic gels were prepared with 52.5% rat tail collagen I (BD Biosciences, 354236), 17.5% Matrigel (BD Biosciences, 354234), 10% FBS (Gibco, 10500), and 10% 10× DMEM (Sigma, D2429). Gel mixture was neutralized by adding 1 M NaOH (Sigma, S8045), and then, $5 \times 10^5$ cells were embedded in gels in pertinent media (10% of total gel volume). 1 ml gel mixture was aliquoted per well of a 24-well plate. Gels were set at 1 h at 37°C then maintained with the pertinent media for 3 days (when contraction is observed). PSCs were killed by incubating the gels with 400 μg/ml hygromycin (Life Technologies, 10687-010) containing culture media for 48 h. After that, gels were washed with PBS for 45 min 3 times. Then, $2.5 \times 10^5$ AsPc1 cells (2:1 ratio for PSC: Cancer cell) were seeded on top of the gels and incubated overnight. After that gels were lifted to an air–liquid interface on top of rat tail collagen I-coated nylon membranes (100 μm pore size, Millipore, NY1H02500) placed on stainless steel grids and fed from beneath

for 10 days with 10% FBS containing RPMI (Sigma, R8758). Then, gels were harvested, fixed overnight with formalin (Sigma, HT501128-4L), and embedded in paraffin (Fisher, 12624077). 4 μm sections were cut and stained for H&E. Images were captured by using AE2000 binocular microscope (Motic) at 20× magnification with Leica Application Suite 3.6 software. The number of invading cell cohorts was counted using ImageJ (NIH, 1.47v). Briefly, bright field H&E images were changed to 8-bit type then converted into binary. The holes were filled, and the invasion index is calculated as 1 − (non-invading cell area/total cell area). The invading cohorts were counted, and the total area was calculated by restricting the size analysis to the size interval of cohorts and circularity to 0–1. Total number of invading particles per field was presented as one data point.

## G-LISA assay for RhoA

The intracellular amounts of total RhoA and RhoA-GTP were determined by using the total RhoA ELISA and G protein-linked (G-LISA) assays (Cytoskeleton, Inc., Denver, CO, USA) according to the manufacturer's instructions. Briefly, cells were washed with cold PBS and homogenized gently in ice-cold lysis buffer. 20 μl was removed for protein quantification in order to adjust sample concentration to 0.5 mg/ml. After adding an equal volume of binding buffer, triplicate assays were performed using 1.5 μg protein per well. Samples were incubated for 30 min and then washed three times with washing buffer. Antigen-presenting buffer was added for 2 min before removal; samples were then incubated with 1:250 dilution of anti-RhoA antibody at room temperature for 45 min, washed three times, and incubated with secondary antibodies for another 45 min. HRP detection reagent was added and signal was read by measuring absorbance at 490 nm using a microplate spectrometer.

## RhoA rescue and functional assays

Pancreatic stellate cells were treated with 1 μM Tam for 10 days, then transfected with 2 μg RhoA constitutively active plasmid (pcDNA3-EGFP-RhoA-Q63L, a gift from Gary Bokoch, Addgene plasmid # 12968), for 4 h by using JetPRIME reagent (1:3 DNA:-jetPRIME ratio (w/v)) and JetPRIME buffer (Polyplus, 114-15). Mock transfection was done by using JetPRIME reagent and buffer only (i.e., without DNA), and the cells were otherwise treated the same way as the transfection group. Functional assays were done 48 hours after transfection. To study the effect of active RhoA over-expression on ECM remodeling, active RhoA over-expressing tamoxifen-treated PSCs and mock transfection group were trypsinized and 500,000 cells were embedded in 80 μl collagen I/Matrigel mixture gels (4.5 mg/ml and 2 mg/ml final concentration, respectively). After 1-h incubation at 37°C on 2% BSA-treated wells of 96-well plate, gels were covered media and left to be remodeled 3 days at 37°C. Gel contraction was calculated as % reduction in the gel surface area.

## Migration assay

Cells were cultured on 35-mm glass-bottomed dishes pre-coated with 10 μg/ml fibronectin and grown to a confluence of 95–100% in

culture media DMEM with 2% FBS containing DMEM. Upon reaching confluence, Tam treatment was applied to the treated population for 10 days prior to scratch assay measurements. A linear scratch was applied to the cell monolayer with a sterile 100-μl plastic pipette tip. Cellular debris was removed from the dish through a wash with DMEM prior to measurement. Scratch assays were kept at 37°C and images taken along the length of the scratch were obtained with phase contrast microscopy with a 10× objective. Images were taken at time intervals of 0 and 24 h. Images were analyzed in a custom program (MATLAB) to detect the cell free area in the scratch and the percentage change was calculated to quantify the wound closure.

## Macrophage spreading

Measurements of the time-dependent spreading of macrophages were conducted on glass-bottom Petri dishes (Maktek) coated with human plasma fibronectin (10 μg/ml; Sigma) and incubated at 37°C. Images of the cells were obtained with an inverted microscope (Eclipse Ti; Nikon) in DIC mode with the samples held at 37°C. Images were obtained with a sCMOS camera every 5 min using a × 20 (0.4 numerical aperture (NA), air; Nikon) objective until noticeable cell spreading had stopped. The cell area was calculated using the imaging software (NIS elements; Nikon) by selecting the perimeter of the cell in each frame allowing the cell area to be tracked with time.

## Macrophage attachment to PAA gels

Double rigidity PAA substrates were produced as follows: Soft, 1 kPa PAA gel was prepared by mixing 459.1 μl PBS, 34.9 μl acrylamide/bisacrylamide (29:1), 1 μl TEMED, and 2.5 μl 10% APS (all from Sigma Aldrich). Stiff, 25 kPa PAA gel was prepared with addition of 2.5 μl yellow-green 0.2 μm FluoSpheres carboxylate (Molecular Probes, USA) so as to distinguish the boundary between rigidities, 378.7 μl PBS, 125.3 μl acrylamide/bisacrylamide, 1 μl TEMED, and 2.5 μl 10% APS. 8 μl single drops of both polymer solution was then placed within 1 mm distance on 3-(trimethoxysilyl)propyl methacrylate-treated glass-bottom fluorodishes. In order to form the flat gel surface with a rigidity boundary, dichlorodimethylsilane (Sigma-Aldrich, USA)-treated glass coverslip was placed on top. Gels were incubated for 45 min to allow polymerization before gentle removal of the dichlorodimethylsilane-treated coverslip. Gels were then sterilized under 2 × 30 min of UV light. To allow cell attachment to gels, 50 μl sulfo-SANPAH (Sigma-Aldrich, USA) solution (1 mg/ml in PBS) was used to covalently bind native human fibronectin (Gibco, USA) to gel surface. Sulfo-SANPAH-coated gels were exposed to 2 × 5 min UV radiation. After 2 × 5 min PBS wash, 50 μl of fibronectin solution (10 μl fibronectin/1 ml PBS) was added on the gel surface and incubated in 4°C overnight. Excess fibronectin was then removed with gentle PBS washing. Macrophage cells on double rigidity gels were analyzed using Nikon Ti-e microscope 16 h after seeding. Gels with cells were transferred to a microscope culture chamber (37°C, 5% $CO_2$) and the images from both stiff (labeled with FluoSpheres) and soft side were taken using NIS Elements software. Cell number, area, and roundness were quantified using Fiji.

## Macrophage transwell assay

Transwell culture insert with 8-μm pore polycarbonate membrane (CLS3422-48EA, Corning®, UK) was used. The bottom of the multi-wells was treated with Matrigel solution used as chemoattractant at a final concentration of 2.2 mg/ml in 10× DMEM (D2429), and the inner part of the culture inserts was treated with rat tail collagen type-I solution at a final concentration of 4.6 mg/ml in 10× DMEM before the culture inserts were plated. Both Matrigel and collagen were incubated at 37°C for 1 h. Murine macrophage (RAW 264.7) were cultured in Dulbecco's modified Eagle's medium–high glucose medium (DMEM, D6429, Sigma-Aldrich, UK). Control and treated macrophages ($1 \times 10^5$ per assays and per condition) were suspended in 100 μl serum free clear media and pipetted into the collagen-rich transwell without touching the membrane or introducing air bubbles. Macrophages were incubated for 16 h. Afterward, each culture insert was rinsed and transferred to a new reservoir containing 100% of absolute ethanol (VWR) to fix the cells in the polycarbonate membrane and stained with 0.1% crystal violet using the standard protocol. Images of the bottom side of the membranes were taken with Motic AE31 trinocular inverted microscope by Motic Images Plus 2.0 software using 20× objective. Crystal violet-stained cells were analyzed in Fiji. The number of invaded cells per each transwell was quantified by dividing the number of cells in each region of interest by the area of the microscope viewing field and then multiplied by the entire area of the Transwell insert. Invasion assays were performed in triplicate.

## Statistical analysis

All statistical analyses were conducted with the Prism graphical software (GraphPad, Software). Data were generated from multiple repeats of different biological experiments in order to obtain the mean values and standard errors (s.e.m) displayed throughout. *P*-values have been obtained through *t*-tests on paired or unpaired samples with parametric tests used for data with a normal distribution and non-parametric tests conducted via the Mann–Whitney test where data had a skewed distribution. Significance for the *t*-tests was set at $P < 0.05$ where graphs show significance through symbols (*$P < 0.05$; **$P < 0.01$; ***$P < 0.001$). For experiments with more than two groups, ANOVA and the indicated post hoc test were used.

**Expanded View** for this article is available online.

## Acknowledgements
We are very grateful to Saadia Karim and Jennifer Morton for providing us with pancreatic tissues from KPC mice and Francesco Di Maggio for help in implementing the initial work with pancreatic stellate cells in the group. We are also grateful to Tyler Lieberthal and Alistair Rice for critical reading of the manuscript and to Tyler Lieberthal for help with the model figure. This work was supported by the European Research Council (ERC grant 282051 - Force-Regulation), and the Biotechnology and Biological Sciences Research Council (BBSRC grant BB/N018532/1).

## Author contributions
EC carried out GLISA, ELISA, RhoA rescue assays and developed the 3D gel contraction methodology; EC and MS performed IHC, IF, and WB experiments

and organotypic assays; EC and BR carried out AFM studies and cellular experiments for migration and YAP activation; EC and AC conducted mechano-sensing experiments; EC and DL performed *in vitro* macrophage invasion assays, qPCR and IF. SDT performed WB assays and analyzed data under the supervision of DAL. SDT contributed with manuscript preparation. AERH analyzed data and supervised research; EC and AERH conceived the idea for this project and wrote the manuscript with significant inputs from all authors.

## Conflict of interest

The authors declare that they have no conflict of interest.

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
