## [Review Process File · EMBO Reports]

GPER is a mechanoregulator of pancreatic stellate cells and the tumor microenvironment

Ernesto Cortes, Muge Sarper, Benjamin Robinson, Dariusz Lachowski, Antonios Chronopoulos, Stephen D. Thorpe, David A Lee and Armando E. del Río Hernández

Review timeline:	Submission date:	9 June 2018
	Editorial Decision:	16 July 2018
	Revision received:	12 October 2018
	Accepted:	23 October 2018

Editor: Achim Breiling

Transaction Report:

1st Editorial Decision

16 July 2018

Thank you for the submission of your manuscript to our editorial offices. We have now received the reports from the referees that were asked to evaluate your study (you will find enclosed below). These were the same four referees that have assessed your related submission, that has been rejected by The EMBO Journal before.

As you will see, the referees now mostly support publication of your manuscript in EMBO reports (and of the accompanying study submitted back-to-back - see also the decision letter for EMBOR-2018-46557). However, the referees have still several suggestions to improve the paper, and some minor concerns, that we ask you to address in a final revised version of the manuscript. As also indicated by referee #1, we think that the paper should be published as Short Report, and needs to be formatted accordingly (see below). Please also address the two major points by referee #4 in your point-by-point-response, and/or with appropriate changes in the manuscript text. Please also discuss your data in the light of the accompanying paper, and cite the other work.

I also have the following editorial requests that also need to be addressed:

- The abstract is currently too long. Please provide an abstract with not more than 175 words.
- Please add a short running title (less than 40 characters w/o spaces) to the title page.
- As mentioned above, we would like to publish the paper as Scientific Report. For a Scientific Report we require that results and discussion sections are combined in a single chapter called "Results & Discussion". Please do that for your manuscript.

For more details please refer to our guide to authors:
<http://embor.embopress.org/authorguide#manuscriptpreparation>

- For a short report, we would require 5 main figures. I think it should be no problem to fit the main data you have now into 5 main figures and up to 5 EV figures. The Expanded View format, which will be displayed in the main HTML of the paper in a collapsible format, has replaced the Supplementary information. You can submit up to 5 images as Expanded View. Please name these

figures following the nomenclature Figure EV1, Figure EV2 etc. The figure legend for these should be included in the main manuscript document file in a section called Expanded View Figure Legends after the main Figure Legends section. Please update all the figure callouts in the manuscript text.

- Additional Supplementary material should be supplied as a single pdf labeled Appendix. The Appendix includes a table of content on the first page, all figures and their legends, and should include page numbers. Please follow the nomenclature Appendix Figure Sx throughout the text and also label the figures according to this nomenclature.

See also our guide for figure preparation:

http://www.embopress.org/sites/default/files/EMBOPress_Figure_Guidelines_061115.pdf

- All materials and methods should be included in the main manuscript file.

- Please provide all scale bars in Fig. 1E without text on them.

- It seem there is presently no callout for Fig. 7. Please check.

- Please indicate in the author checklist in field D10 that you comply with the ARRIVE guidelines, and re-submit the modified form. See also our guidelines for the use of living organisms, and the respective reporting guidelines: <http://embor.embopress.org/authorguide#livingorganisms>

- Please also format the references according to EMBO reports style. See: <http://embor.embopress.org/authorguide#referencesformat>

- We now strongly encourage the publication of original source data with the aim of making primary data more accessible and transparent to the reader. The source data will be published in a separate source data file online along with the accepted manuscript and will be linked to the relevant figure. If you would like to use this opportunity, please submit the source data (for example scans of entire gels or blots, data points of graphs in an excel sheet, additional images, etc.) of your key experiments together with the revised manuscript. Please include size markers for scans of entire gels, label the scans with figure and panel number, and send one PDF file per figure.

- a Microsoft Word file (.doc) of the revised manuscript text
- a letter detailing your responses to the final referee comments in Word format (.doc)
- editable TIFF or EPS-formatted figure files (main figures and EV figures) in high resolution
- the revised Appendix

In addition I would need from you:

- a short, two-sentence summary of the manuscript
- two to three bullet points highlighting the key findings of your study
- a schematic summary figure (in jpeg or tiff format with the exact width of 550 pixels and a height of about 400 pixels) that can be used as visual synopsis on our website.

I look forward to seeing the final revised version of your manuscript when it is ready. Please let me know if you have questions or comments regarding the revision.

REFeree REPORTS

Referee #1:

In the paper GPER is a mechano-regulator of pancreatic stellate cells and the tumor microenvironment, the authors demonstrate that tamoxifen treatment of PSCs in vitro and in vivo inhibits/reverts the activation of PSCs. The outcome is lower ECM remodeling due to lower actomyosin contraction and YAP signaling. The authors demonstrate that the tamoxifen works through the GPER and that this should be considered as a novel anti-stromal therapy in PDACs.

The paper is acceptable for publication in EMBO Report as a short communication although some minor correction would improve the quality of the paper.

The general comments to many of the experiments is how much of the difference can be accounted to the fact that tamoxifen treated PSCs proliferate remarkably less to control PSCs? This reviewer appreciates the differences in all single cell analyses, but it would be important to have an estimation to what level the lower proliferation contributes to all the inhibitory functions of multiple cells assays like i.e. contraction assays, invasion assays, and wound healing assay? The second paper seem to suggest that the effect is mostly prominent from day 5.

As tamoxifen is normally given to women with ER-driven tumours, it would be nice to know if the tamoxifen effect through GPER works equally well on female and male mice? Do the authors have data on that? And could the authors please specify which mice were used for the study!

The authors could potentially rearrange to figures a bit in order to make the study more streamline (they jump a bit around). Some of the figures could easily be fused and the paper could be made into a short communication (3 main figures + the model figure). For instance, it is nice to see that total MLC is not changed by tamoxifen (as YAP activities has been shown to regulate total levels of MLC), but the data could easily be move to supplementary figures. The authors could represent the data as pMLC/MLC.

- Figure 2 about the macrophages should be placed in the Supplementary as the paper is about PSCs, not macrophages.
- Figure 3 and 4 could be fused.
- Figure 5 and 6 could be fused.

Specific comments:

1. How was the CD204 stain quantified? The stain looks brown everywhere with very high background.
2. Please correct the title in Fig 2a. to '... on fibronectin-coated glass'.
3. Page 8: change the sentence 'However, inhibition was suppressed when the GPER antagonist was used (Fig. 2G-I)' to ' However, inhibition was alleviated when the GPER antagonist was used (Fig. 2G-I).
4. Figure 5a, these images are fare from convincing. Why is there DAPI staining in the cytosol and outside the cells? Are the cells mycoplasma free? The dapi staining is lower in TAM treated cells suggesting that the images are taken at a different z-plane with less nucleus being imaged in TAM treated cells. How many times has this experiment being conducted for the quantification?
5. Suppl. Figure 3. Could they not show a proper proliferation curve over days instead of Ki67 and apoptosis? This actually shown in the accompanying paper!
6. Suppl fig 9 third panel. The cells have obviously migrated more than 2-3 microns. Do they mean 3 mm? Please correct.
7. Suppl fig 10. Again their DAPI staining is awful - the same is the YAP staining. How can they quantify (Suppl 10b) anything with this quality? Is this really reliable?

Referee #2:

The study by Cortes et al addresses how tamoxifen affects contraction of pancreatic stellate cells and macrophage motility and recruitment within pancreatic tumors. The study is overall well performed and the findings are intriguing. However, I have a few minor points to address.

1. The term 'deactivation' is used in a number of places in the text (for example abstract and headlines). This term is rather unspecific and should be replaced with a more specific term that

describes better what occurs in response to tamoxifen.

2. In Figure 7, it is not clear to this reviewer whether including quiescent PSC is relevant, since PSC proliferation was not analyzed. The figure legend lacks a description of what the authors want to present in the model.

3. The direct tamoxifen effect on macrophages should be acknowledged better, for example in the discussion and abstract. It appears as if the authors conclude that all the effect observed in the tumor is caused by changes in PSC contraction and biomechanics. However, based on the results (for example Figure 2) the model is likely to be more complicated.

Referee #3:

Overall, these two submissions have a relatively tight focus on the 'softening effect' of Tamoxifen on PDAC tissues. Conventional expression profiling is used to quantify the loss of contractility and matrix components at protein and transcript levels, giving a consistent picture. Overall, regulatory connections are not always clear, and some thoroughness seems lacking in the main figures.

In the manuscript "GPER is a mechano-regulator...", my only concerns are for Fig.5:

1. Fig.5A,B analysis of YAP is unconvincing.
2. Total YAP levels are also needed for Fig.5D,E.
3. Does CTGF control collagen levels?

Referee #4:

Comments for manuscript #1 "GPER is a mechano-regulator of pancreatic stellate cells and the tumor microenvironment": Conclusions/possible applications of these findings are too broad and not supported by the data or the point-by-point response to the reviewers in the following points:

a) The authors argue that their data highlight the need for caution in using tamoxifen inducible Cre mice models. However, Reviewer #1 -expert in PDAC models and Tumor microenvironment- points out that "The caution for other investigators the authors want to raise may also not be necessary. Most investigators use tamoxifen briefly (a few days) to get a disease process started that progresses without tamoxifen. It can take weeks to months for the disease they want to study to develop in these models. So, at the time the tamoxifen is used there is no cancer and by the time a cancer forms there is no tamoxifen". The authors argued that whereas this is true for Cre systems that trigger acute conditions, there some cases where long-term exposure of tamoxifen is needed to recreate chronic conditions, so their findings would be relevant for the latter experimental setups. They also argue that their findings are also relevant for those cases when the experimental setup requires short exposures to tamoxifen but the output is assessed immediately after treatment. In light of the authors' answers, they should re-write their introduction and discussion to state that their findings would be only applicable to these specific Cre inducible experimental setups.

b) Reviewer #1 points out that the doses of tamoxifen tested in this study are 200-1000 times higher than those used clinically when tamoxifen is given. The authors' arguments for choosing these concentrations are: 1) They used these concentrations "to match the studies using tamoxifen in genetically modified mouse models". However, their conclusion regarding the application of these findings to inducible Cre mice is only secondary in this manuscript and, as explained in my point above, should even be narrowed because the broader implications that the authors want to raise are not supported by their data. Therefore, there is no clear link between this argument and the findings presented in this paper. 2) The authors argue that 2mg tamoxifen represented a 100mg/kg dose

which leads to serum tamoxifen concentrations of 0.5 - 1 μ M "which corresponds to the serum concentrations found in humans after administration of clinical doses of 20 mg/day" according to DeGregorio et al, 1987 - PMID: 3690805. However, the referenced paper finds that 100mg/kg (the highest concentration tested in the cited paper) produces an average serum concentrations of 0.51 μ M (0.20-0.72). Therefore, on one hand the authors should correct their statement in the current paper where they say that based on DeGregorio et al. paper, 2mg (100mg/kg) produces a 0.5-1 μ M serum concentration, which is incorrect. On the other hand, the conclusion of the referenced paper is that "Daily s.c. injections of 1 mg or i.p. injections of 25-100 mg/kg achieved clinically relevant serum tamoxifen levels and can therefore be used to study the effects of tamoxifen on human breast cancer". This indicates that the lowest dose used in the present manuscript is already at the highest level recommended to use in the cited paper, which supports the concern the chosen dose and what would be the clinical relevance of using an even higher concentration of 5mg (250mg/kg) when the 100mg/kg is already an example of high-dose tamoxifen administration. Therefore, the authors failed to provide a compelling argument for why these concentrations would be clinically relevant. Since the lowest concentration used in this manuscript seems to represent a high-dose scenario in patients, their conclusions should be adjusted to explicitly state that this paper is focusing in high-dose tamoxifen administration and their discussion regarding the clinical relevance of this study could be avoided.

1st Revision - authors' response

12 October 2018

Referee #1:

In the paper GPER is a mechano-regulator of pancreatic stellate cells and the tumor microenvironment, the authors demonstrate that tamoxifen treatment of PSCs in vitro and in vivo inhibits/reverts the activation of PSCs. The outcome is lower ECM remodeling due to lower actomyosin contraction and YAP signaling. The authors demonstrate that the tamoxifen works through the GPER and that this should be considered as a novel anti-stromal therapy in PDACs.

The paper is acceptable for publication in EMBO Report as a short communication although some minor correction would improve the quality of the paper.

The general comments to many of the experiments is how much of the difference can be accounted to the fact that tamoxifen treated PSCs proliferate remarkably less to control PSCs? This reviewer appreciates the differences in all single cell analyses, but it would be important to have an estimation to what level the lower proliferation contributes to all the inhibitory functions of multiple cells assays like i.e. contraction assays, invasion assays, and wound healing assay? The second paper seem to suggest that the effect is mostly prominent from day 5.

Authors: Assays that measured a bulk cell response were conducted within 72 h [gel contraction assays – 72h, ECM remodelling by PSCs in cancer cell invasion assays – 72h, wound healing assays – 24h]. At the beginning of the experimental setup for each assay, equal numbers of cells have been used for the control and tamoxifen treated groups. We monitored the effect of tamoxifen treatment on PSC proliferation for ten days. There is no significant difference in the proliferation rate of control and tamoxifen treated PSCs at 72h. Significant differences between the control and tamoxifen groups are observable after the 5th day. This data is included in the accompanying paper (Figure EV2).

As tamoxifen is normally given to women with ER-driven tumours, it would be nice to know if the tamoxifen effect through GPER works equally well on female and male mice? Do the authors have data on that? And could the authors please specify which mice were used for the study!

Authors: We used a total of 15 mice in our study. **Control:** 3 males/ 2 females; **2mg:** 3 males/ 3 females; and **5mg:** 2 males/2 females. This data has been included in the method section. We do not have data about the differential effect of tamoxifen via GPER depending on sex.

Circulating estrogens have high affinity for GPER receptors and can modulate human physiology independently of the well-studied genomic effects on the canonical estrogen response elements.

Although endogenous estrogen is mainly derived from the ovaries in premenopausal women and mostly regarded as a female hormone [PMID: 16511588]; in both males and females estrogen is produced in other tissues, such as adipose tissues and arteries [PMID: 21844907 and PMID: 11248122]. It is well documented that women with decreased circulating estrogen develop hypertension, diabetes, and obesity [PMID: 19818376 and PMID: 25022814], and the molecular mechanisms by which GPER is involved in these diseases have started to emerge [PMID: 27803283]. Further studies on the effect of circulating estrogen in GPER activation will broaden our basic understanding of several age-derived diseases in males and females.

The authors could potentially rearrange to figures a bit in order to make the study more streamline (they jump a bit around). Some of the figures could easily be fused and the paper could be made into a short communication (3 main figures + the model figure). For instance, it is nice to see that total MLC is not changed by tamoxifen (as YAP activities has been shown to regulate total levels of MLC), but the data could easily be move to supplementary figures. The authors could represent the data as pMLC/MLC.

- Figure 2 about the macrophages should be placed in the Supplementary as the paper is about PSCs, not macrophages.
- Figure 3 and 4 could be fused.
- Figure 5 and 6 could be fused.

Authors: We have rearranged the figures to fit all data into a short report format. Figure 2 is included in supplementary information, and the rest of figures have been arranged to fit within this format.

Specific comments:

1. How was the CD204 stain quantified? The stain looks brown everywhere with very high background.

Authors: This staining was repeated to confirm our previous observations. New images with and new quantification are provided. We quantified the percentage of nuclei included in the CD204 stained areas for each condition. This percentage was normalised to the control condition. The values obtained from the different regions of the same tissue sections were averaged and treated as an experimental replicate. The final result was calculated for n=4 animals per condition. The figure below represents one of the new CD204 images. Brown is CD204, blue is haematoxylin.

2. Please correct the title in Fig 2a. to '... on fibronectin-coated glass'.

Authors: This has been changed in the current version. Many thanks.

3. Page 8: change the sentence 'However, inhibition was suppressed when the GPER antagonist was used (Fig. 2G-I)' to ' However, inhibition was alleviated when the GPER antagonist was used (Fig. 2G-I).

Authors: This has been changed in the current version.

4. Figure 5a, these images are fare from convincing. Why is there DAPI staining in the cytosol and outside the cells? Are the cells mycoplasma free? The dapi staining is lower in TAM treated cells suggesting that the images are taken at a different z-plane with less nucleus being imaged in TAM treated cells. How many times has this experiment being conducted for the quantification?

Authors: Many thanks for this comment. We conducted a mycoplasma test and confirmed that cells were not contaminated. We performed an independent set of experiments (4 biological samples measured in 3 different experiments). New images are shown below.

5. Suppl. Figure 3. Could they not show a proper proliferation curve over days instead of Ki67 and apoptosis? This actually shown in the accompanying paper!

Authors: We conducted Ki67 and Cc3 caspase assays to quantify the direct effect of tamoxifen in the number of proliferating cells and cells dying through apoptosis separately. We observed a decrease in proliferation and increased apoptosis after tamoxifen treatment. This suggests that the

curve of percentage of cells over days should increase to a lower extent in the tamoxifen treated cells compared to control. The single cells analyses are not affected by this effect and the cell attachment to PAA and cell invasion analyses were conducted during 16h. Based on our studies in pancreatic stellate cells (accompanying paper), the effect of tamoxifen in proliferation is present at day 5, which points to the notion that tamoxifen affects proliferation through genomic signalling. This requires days to manifest and therefore would not interfere with our observations within 16h after cell seeding.

6. Suppl fig 9 third panel. The cells have obviously migrated more than 2-3 microns. Do they mean 3 mm? Please correct.

Authors: Many thanks for spotting this. It has been corrected.

7. Suppl fig 10. Again their DAPI staining is awful - the same is the YAP staining. How can they quantify (Suppl 10b) anything with this quality? Is this really reliable?

Authors: We provide a new set of images that better represent the DAPI and YAP staining for our immunofluorescent experiments. Images are presented below. This is Appendix Figure S10 in the current version. The white arrows indicate nuclear YAP localisation (in the control group), and nuclei devoid of YAP (in the tamoxifen treated group).

How quantification was done: Images of the nuclei stained with DAPI were thresholded, resulting in a binary mask of the nuclei. This mask was overlaid on the YAP immunofluorescence image and the mean fluorescent intensity was measured giving the average YAP expression in the nucleus. Subsequently, the binary mask was created from the YAP immunofluorescence staining. To obtain the cytoplasmic YAP expression the nuclear mask was subtracted from the YAP mask. The ratio between the first and the second value gave the nuclear to cytoplasmic YAP expression ratio. The values obtained from the different regions of the same tissue sections were averaged and treated as an experimental replicate. The final results were calculated for n=3 animals per condition.

Referee #2:

The study by Cortes et al addresses how tamoxifen affects contraction of pancreatic stellate cells and macrophage motility and recruitment within pancreatic tumors. The study is overall well performed and the findings are intriguing. However, I have a few minor points to address.

1. The term 'deactivation' is used in a number of places in the text (for example abstract and headlines). This term is rather unspecific and should be replaced with a more specific term that describes better what occurs in response to tamoxifen.

Authors: The term 'deactivation' has been replaced with 'Tamoxifen inhibits the myofibroblastic differentiation'. The term 'myofibroblastic differentiation' illustrates the process by which pancreatic stellate cells get activated and adopt a contractile phenotype (expressing high levels of aSMA and the intermediate filament vimentin). Many thanks for this comment.

In the text: The processes involving myofibroblast differentiation are presented in the *introduction section*: 'An integral feature of PSCs in PDAC is their transition to an activated state whereby they lose their cytoplasmic vesicles and adopt a myofibroblast-like contractile phenotype expressing high levels of alpha smooth muscle actin (aSMA) [2]'.

2. In Figure 7, it is not clear to this reviewer whether including quiescent PSC is relevant, since PSC proliferation was not analyzed. The figure legend lacks a description of what the authors want to present in the model.

Authors: The former figure 7 (model) is inserted in Figure 5 (last panel - G) in the current version. The term 'quiescent' has been replaced with 'inactive PSC'. A description of the model is included in the figure legend where the meaning of the term 'inactive PSC' is provided.

In the figure legend: (G) Illustration of the effect of tamoxifen on GPER activation at the cellular level (left panel) and tissue level (right panel). In PSCs, tamoxifen suppresses the activation of YAP and MLC-2 (by phosphorylation to pMLC-2) via the axis GPER/RhoA. This inhibits mechanosensing and the ability to apply endogenous forces, which are required to maintain the myofibroblastic phenotype in PSCs. Consequently, PSCs adopt a mechanically inactive state (not myofibroblast like cell). Pancreatic tissues from KPC mice treated with tamoxifen have reduced tissue stiffness and desmoplastic reaction (decreased collagen deposition). The expression of the myofibroblast marker α -SMA is reduced consistent with the mechanical inactivation of PSCs. The recruitment of macrophages, their M2 polarization, and the invasion of cancer cells, are also reduced in these tissues.

3. The direct tamoxifen effect on macrophages should be acknowledged better, for example in the discussion and abstract. It appears as if the authors conclude that all the effect observed in the tumor is caused by changes in PSC contraction and biomechanics. However, based on the results (for example Figure 2) the model is likely to be more complicated.

Authors: This comment has been addressed in the revised version.

In the abstract: Tamoxifen also reduces the recruitment and polarization to the M2 phenotype of tumor associated macrophages.

In the discussion section: Tamoxifen reduces tissue stiffness, collagen deposition, collagen fiber thickness, and α SMA expression in PDAC mouse models. Concurrently, tamoxifen impedes the recruitment of tumor associated macrophages and in particular their polarization towards the M2 phenotype that are highly associated with invasion and metastasis in PDAC [22, 26]. A summary of these interactions is shown in Figure 6G.

Referee #3:

Overall, these two submissions have a relatively tight focus on the 'softening effect' of Tamoxifen on PDAC tissues. Conventional expression profiling is used to quantify the loss of contractility and matrix components at protein and transcript levels, giving a consistent picture. Overall, regulatory connections are not always clear, and some thoroughness seems lacking in the main figures.

In the manuscript "GPER is a mechano-regulator...", my only concerns are for Fig.5:

1. Fig.5A,B analysis of YAP is unconvincing.

Authors: We appreciate this Reviewer's comment that resonates with the question from Reviewer 1 point 4. New images and quantification have been provided. New images are shown below. This is presented in Figure 4 panel A and B of the current version.

2. Total YAP levels are also needed for Fig.5D,E.

Authors: We used Western blot to determine the total levels of proteins, total YAP and YAP phosphorylated in position Ser127. The results are presented in Figure 4 D-E.

In the text: Both total YAP and pYAP Ser127 were reduced in PSCs in response to tamoxifen by approximately 33% and 22%, respectively. However, pYAP Ser127 was reduced to a lesser extent such that the ratio between pYAP and total YAP actually increased by approximately 17% in the tamoxifen treated PSCs.

3. Does CTGF control collagen levels?

Authors: Many thanks for raising this question. CTGF has been shown to promote collagen I expression in lung fibroblasts, acting through c-Jun phosphorylation and recruitment of c-Jun and c-Fos to the collagen I promoter (PMID: 23906792). Additionally, a CTGF response element has been detected in the promoter for collagen I in skin fibroblasts, which allows CTGF-mediated upregulation of collagen I expression (PMID: 10942593). Knockdown of CTGF by RNA interference has also been shown to reduce collagen I expression in human cardiac fibroblasts (PMID: 29287092).

These studies collectively indicate that the decrease in CTGF expression we see in the myofibroblast-like pancreatic stellate cells following tamoxifen treatment is highly likely to directly impact collagen expression levels.

Referee #4:

Comments for manuscript #1 "GPER is a mechano-regulator of pancreatic stellate cells and the tumor microenvironment": Conclusions/possible applications of these findings are too broad and not supported by the data or the point-by-point response to the reviewers in the following points:

a) The authors argue that their data highlight the need for caution in using tamoxifen inducible Cre mice models. However, Reviewer #1 -expert in PDAC models and Tumor microenvironment- points out that "The caution for other investigators the authors want to raise may also not be necessary. Most investigators use tamoxifen briefly (a few days) to get a disease process started that progresses without tamoxifen. It can take weeks to months for the disease they want to study to develop in these models. So, at the time the tamoxifen is used there is no cancer and by the time a cancer forms there is no tamoxifen". The authors argued that whereas this is true for Cre systems that trigger acute conditions, there some cases where long-term exposure of tamoxifen is needed to recreate chronic conditions, so their findings would be relevant for the latter experimental setups. They also argue that their findings are also relevant for those cases when the experimental setup requires short exposures to tamoxifen but the output is assessed immediately after treatment. In light of the authors' answers, they should re-write their introduction and discussion to state that their findings would be only applicable to these specific Cre inducible experimental setups.

Authors: The introduction was changed in the revised version to address this comment.

Before: Intriguingly, tamoxifen is widely used to induce the expression of specific phenotypes in conditional somatic mouse mutants (experimental mice for inducible gene knockouts) [14], and its administration may alter the biomechanical homeostasis and immune responses of the tissue under study. This highlights the need for caution in using these tamoxifen inducible Cre mice models, particularly in studies that monitor processes that may be tightly regulated by mechanical cues.

Revised text: Intriguingly, tamoxifen is widely used to induce the expression of specific phenotypes in conditional somatic mouse mutants (experimental mice for inducible gene knockouts) [14], and its administration may alter the biomechanical homeostasis and immune responses of the tissue under study. This highlights the need for caution in using these tamoxifen inducible Cre mice models in the cases where long-term exposure to tamoxifen is needed to resemble chronic conditions, or when the output is assessed immediately after tamoxifen treatment.

In the discussion section: *This section has been rewritten and the discussion about the Cre inducible experimental setups has been removed.*

b) Reviewer #1 points out that the doses of tamoxifen tested in this study are 200-1000 times higher than those used clinically when tamoxifen is given. The authors' arguments for choosing these concentrations are: 1) They used these concentrations "to match the studies using tamoxifen in genetically modified mouse models". However, their conclusion regarding the application of these findings to inducible Cre mice is only secondary in this manuscript and, as explained in my point above, should even be narrowed because the broader implications that the authors want to raise are not supported by their data. Therefore, there is no clear link between this argument and the findings presented in this paper. 2) The authors argue that 2mg tamoxifen represented a 100mg/kg dose which leads to serum tamoxifen concentrations of 0.5 - 1uM "which corresponds to the serum concentrations found in humans after administration of clinical doses of 20 mg/day" according to DeGregorio et al, 1987 - PMID: 3690805. However, the referenced paper finds that 100mg/kg (the

highest concentration tested in the cited paper) produces an average serum concentrations of 0.51 uM (0.20-0.72). Therefore, on one hand the authors should correct their statement in the current paper where they say that based on DeGregorio et al. paper, 2mg (100mg/kg) produces a 0.5-1uM serum concentration, which is incorrect. On the other hand, the conclusion of the referenced paper is that "Daily s.c. injections of 1 mg or i.p. injections of 25-100 mg/kg achieved clinically relevant serum tamoxifen levels and can therefore be used to study the effects of tamoxifen on human breast cancer". This indicates that the lowest dose used in the present manuscript is already at the highest level recommended to use in the cited paper, which supports the concern the chosen dose and what would be the clinical relevance of using an even higher concentration of 5mg (250mg/kg) when the 100mg/kg is already an example of high-dose tamoxifen administration. Therefore, the authors failed to provide a compelling argument for why these concentrations would be clinically relevant. Since the lowest concentration used in this manuscript seems to represent a high-dose scenario in patients, their conclusions should be adjusted to explicitly state that this paper is focusing in high-dose tamoxifen administration and their discussion regarding the clinical relevance of this study could be avoided.

Authors: The text has been changed according to this reviewer's suggestion.

In the manuscript text:

The following text was removed from the results section:

'The 2 mg dose in mice (100 mg/kg) gives serum tamoxifen concentration from 0.5 to 1 mM, which corresponds to the serum concentrations found in humans after administration of clinical doses of 20 mg/day [18]. The 5mg dose is selected as an example for high-dose tamoxifen administration.'

The following text was added in the Results and Discuss section:

We acknowledge that our in vivo studies focused on high-dose tamoxifen administration, and scaling this dose based on body weight in humans would result in supraphysiologic doses, for which limited safety data exists. Therefore, future studies using lower doses are required for further clinical validation.

YOU MUST COMPLETE ALL CELLS WITH A PINK BACKGROUND ↓
PLEASE NOTE THAT THIS CHECKLIST WILL BE PUBLISHED ALONGSIDE YOUR PAPER

Corresponding Author Name: Armando E. del Río Hernández

Manuscript Number: EMBOR-2018-46556V1